# Cuffless and Touchless Measurement of Blood Pressure from Ballistocardiogram Based on a Body Weight Scale

**DOI:** 10.3390/nu14122552

**Published:** 2022-06-20

**Authors:** Shing-Hong Liu, Bing-Hao Zhang, Wenxi Chen, Chun-Hung Su, Chiun-Li Chin

**Affiliations:** 1Department of Computer Science and Information Engineering, Chaoyang University of Technology, Taichung City 41349, Taiwan; shliu@cyut.edu.tw (S.-H.L.); s11027608@gm.cyut.edu.tw (B.-H.Z.); 2Biomedical Information Engineering Laboratory, The University of Aizu, Aizu-Wakamatsu City 965-8580, Fukushima, Japan; wenxi@u-aizu.ac.jp; 3Institute of Medicine, School of Medicine, Chung-Shan Medical University, Taichung City 40201, Taiwan; such197408@gmail.com; 4Department of Internal Medicine, Chung-Shan Medical University Hospital, Taichung City 40201, Taiwan; 5Department of Medical Informatics, Chung-Shan Medical University, Taichung City 40201, Taiwan

**Keywords:** ballistocardiogram, photoplethysmogram, pulse transit time (PTT), weight scale, blood pressure

## Abstract

Currently, in terms of reducing the infection risk of the COVID-19 virus spreading all over the world, the development of touchless blood pressure (BP) measurement has potential benefits. The pulse transit time (PTT) has a high relation with BP, which can be measured by electrocardiogram (ECG) and photoplethysmogram (PPG). The ballistocardiogram (BCG) reflects the mechanical vibration (or displacement) caused by the heart contraction/relaxation (or heart beating), which can be measured from multiple degrees of the body. The goal of this study is to develop a cuffless and touchless BP-measurement method based on a commercial weight scale combined with a PPG sensor when measuring body weight. The proposed method was that the PTT_BCG-PPGT_ was extracted from the BCG signal measured by a weight scale, and the PPG signal was measured from the PPG probe placed at the toe. Four PTT models were used to estimate BP. The reference method was the PTT_ECG-PPGF_ extracted from the ECG signal and PPG signal measured from the PPG probe placed at the finger. The standard BP was measured by an electronic blood pressure monitor. Twenty subjects were recruited in this study. By the proposed method, the root-mean-square error (*E_RMS_*) of estimated systolic blood pressure (SBP) and diastolic blood pressure (DBP) are 6.7 ± 1.60 mmHg and 4.8 ± 1.47 mmHg, respectively. The correlation coefficients, *r*^2^, of the proposed model for the SBP and DBP are 0.606 ± 0.142 and 0.284 ± 0.166, respectively. The results show that the proposed method can serve for cuffless and touchless BP measurement.

## 1. Introduction

The COVID-19 virus has been spreading all over the world for more than two years and has led to a large population being isolated at home or quarantined in specified spaces. Thus, physical activity of these isolated people is restricted. According to a previous study, physical inactivity causes more than 5 million deaths worldwide, which does great harm to the finances of public health systems [1]. During physical inactivity, body weight increased, and the weekly energy expenditure and quality of life were reduced [2]. Moreover, the main route of COVID-19 virus infection is the spitting infection. Most people accidentally touch the spit of patients and can very easily become infected. Thus, the World Health Organization suggests that people must wash their hands frequently. The other method is to avoid touching public goods. When the physical activities of people are limited, their dietary assessment will be an important issue to control weight. Moreover, a diet rich in fruits, vegetables, low-fat dairy products, fiber and minerals, would produce a potent anti-hypertensive effect [3,4]. Many studies show that there is a positive relationship between being overweight or obese and blood pressure (BP) with the risk of hypertension [5,6]. Thus, daily measurements of weight and blood pressure could monitor the effectiveness of dietary management by the individual. However, for BP measurement, people must touch the cuff bladder, for which it is very hard to perform sterilization. How to improve the convenience and accuracy of touchless measurements of body weight and BP will be a challenge during this pandemic era.

A pulse wave in the aorta caused by the heart contraction will transmit to the peripheral arterioles. The pulse transit time (PTT) is defined and calculated as the time interval between the R wave of electrocardiogram (ECG) as the starting time and the foot point of photoplethysmogram (PPG) as the ending time [7,8,9]. Some studies used the bioimpedance plethysmogram to replace ECG [10,11], or PPG [12,13]. Moreover, the standard definition of PTT is that the blood pressure pulse wave passes from one position to another position. Weltman et al. derived the pulse wave velocity (PWV) utilizing PTT over a known arterial length [14]. The relationship between PWV and BP was quantitatively evaluated by the well-known Bramwell and Hill equation [15]. The cuffless BP measurement based on the PTT has been studied extensively in the past decades. In 2005, Ahlstrom proposed the use of PTT to estimate the continuous systolic BP that compared with the invasive BP. Its results showed that their correlation could be as high as 0.8. Moreover, when the BP has an instantaneous drop, this method could effectively monitor the rapid change in BP [16]. We all know that the BP measured by the oscillometric method [17] represents an average BP in a short period of time. Thus, it is not suitable for continuous BP measurement. In order to develop a wearable BP measurement with the PTT continuously, some studies only measured the time interval from the wrist to finger [11,18,19]. However, their methods were hard to approach the goal of wearable and touchless measurement.

The ballistocardiography (BCG) converts the pulse pressure in the aorta caused by the heart contraction into mechanical vibration that is transmitted to the body by multiple degrees of freedom [20,21]. A mass-damper-spring model was built to represent the vibration transmission in the body, and to predict the BCG waveforms at upper and lower limb locations. A BCG signal can be measured by a MEMS accelerometer placed at the wrist or foot [22], or by a strain gauge sensor placed at a weight scale [23]. Shin et al. used BCG measured by a MEMS accelerometer placed at the wrist to replace ECG for the BP measurement [20]. Martin et al. used a force sensor placed at the foot to acquire the BCG signal to estimate the BP [23]. However, according to the mass-damper-spring model, the damping and stiffness parameters of a weight scale would affect the pattern and phase of the BCG signal.

The pulse wave measured by PPG is easily affected by some factors, such as the subject’s skin, tissue, and light density of LED. The PTT values derived from PPG and ECG depend largely on the placement of PPG sensors [24]. Moreover, the ECG is an electrical signal, but the BCG is a mechanical signal. Therefore, when the BCG was measured from the weight scale, and the PPG was measured from the PPG probe placed at the toe when a subject was standing on a commercial weight scale, the reliability and reproducibility of BP measurement with the PTT measured by BCG and PPG signals need to be explored. Thus, the goal of this study is to develop a cuffless and touchless BP measurement method based on a commercial weight scale combined with a PPG sensor when a person is measuring body weight. The BCG signal was extracted from the strain gauge of a commercial weight scale, and a PPG probe placed at a toe was used to measure the pulse signal. The PTT measured by the ECG and PPG of the finger as the reference method was compared with the PTT by the proposed method. Four PPT models for estimating BP were used to explore the reliability and reproducibility of the proposed method. Twenty subjects were recruited in this study, and were asked to exercise to raise their blood pressures. The results showed that the BP measured by the proposed method was close to the BP measured by the reference method.

## 2. Materials and Methods

In order to use a commercial weight scale to measure BP in real time, we designed a measurement system including a portable acquisition device, ECG, PPG and BCG driving circuits, and a graphic user interface (GUI) on a notebook PC. The portable acquisition device has eight measuring channels, a 3.0 Bluetooth module, and independent dual power supplies from a battery [25]. Figure 1 shows the structure diagram of the measurement system. The four strain gauges (SGs) of a commercial weight scale (HBF-371, Omron, Osaka City, Japan) were used to sense the body weight and BCG signal. Two PPG probes (DS-100A, Nellcor Puritan Bennett, Pleasanton, CA, USA) were placed on the middle finger of the left hand and a toe of the left foot, respectively, which sensed the PPG signals, PPG_F_ and PPG_T_. The Lead I ECG was measured. Four signals, ECG, BCG, PPG_F_ and PPG_T_, were synchronously acquired and transferred via Bluetooth connection to a notebook PC for data visualization and recording. PTT_ECG+PPGF_ values were extracted from ECG and PPG_F_ signals, and PTT_BCG+PPGT_ values were extracted from BCG and PPG_T_ signals. Then, four PTT models were built to estimate BP.

### 2.1. Portable Acquisition Device

The portable acquisition device is a modified version used in the previous study of Liu et al. [25]. The main part of this portable acquisition device was a 16-bit microcontroller (MCU), TI MSP430 F5438A, of which sampling rate was 500 Hz and resolution of analog to digital conversion (ADC) was 12 bits. The main clock rate of MCU was 24 MHz. The BT 3.0 module (BTM-204B) was used and connected with the MCU by a UART port. A 12-byte transmission package included the first two bytes, 066 as the header of a transmission package, and the remaining ten bytes for ADC values from each channel consisting of high byte and low byte. In the power circuit, a chip (BQ24072, Texas Instruments, Dallas City, TX, USA) for charging the battery was employed, a regulator chip (TPS78233, Texas Instruments, Dallas City, TX, USA) provided a regulated DC voltage of 3.3 V and current of 400 mA, and another regulator chip (TPS60400, Texas Instruments, Dallas City, TX, USA) provided a negative DC voltage of −3.3 V and current of 60 mA. The dual power supplies offered the power not only for the portable acquisition device, but also for the related external ECG, PPG, and BCG circuits.

### 2.2. Ballistocardiograph Circuit

The BCG signal is the alternating signal of SGs produced by the pulse pressure. Its energy and signal-noise ratio (SNR) are very low. Figure 2 shows the BCG circuit that includes an instrumentation amplifier (U1), a highpass filter (U2A and U4B), a noninverting amplifier (U2B and U5A), a lowpass filter (U3 and U4A) and a baseline shifter (U5B). The instrumentation amplifier is AD620 (Analog Device, Norwood City, OH, USA), and the operational amplifier is AD082 (Analog Device, Norwood City, OH, USA). Their working voltages are ±3.3 V. All filters are implemented by the Butterworth structure. The cutoff frequencies of the highpass filter and lowpass filter are 0.3 Hz and 10 Hz, respectively. The total gain is 2000. Finally, the baseline of the BCG signal is shifted to 1.5 V.

### 2.3. Photoplethysmograph Circuit

The PPG circuit detects the alternating component in PPG signal produced by the pulsatile blood flow. Figure 3 shows the PPG circuit that includes a highpass filter (U1A), a noninverting amplifier (U1B), a lowpass filter (U2A and U2B) and a baseline shifter (U3). R1 is a pull-up resistor to convert the current signal to the voltage signal. The operational amplifier is AD082 (Analog Device, Norwood City, OH, USA). Their working voltages are also ±3.3 V. All filters are implemented by the Butterworth structure. The cutoff frequencies of highpass filter and lowpass filter are 0.3 Hz and 40 Hz, respectively. The total gain is 100. Finally, the baseline of PPG signal is shifted to 1.5 V.

### 2.4. Electrocardiograph Module

The chip, AD 8232 (Analog Device, Norwood City, OH, USA) was used to measure Lead I ECG, of which working voltage was 3.3 V, and bandwidth and gain were 0.5 to 40 Hz and 1100, respectively. The baseline is set to 1.5 V.

### 2.5. Digital Signal Processing

All measured signals were filtered to remove the wandering noise in baseline and the high frequency noise using a 4th-order Butterworth bandpass filter in which the lower and upper cutoff frequencies were 0.5 Hz and 10 Hz, respectively. In order to reduce to the differences of phase lag among different signals, an 8th-order all-pass filter was designed to equalize the group delay within the passband. Figure 4 shows the ECG (blue), PPG_F_ (red), PPG_T_ (green), BCG (black), differential PPG_F_ (DPPG_F_, pink) and differential PPG_T_ (DPPG_T_, purple). The PTT based on ECG and PPG signals was derived from the R wave of ECG to the foot point of PPG as PTT1 and from the R wave of ECG to main peak of differential PPG as PTT2 [26]. The PTT measured by BCG and PPG signals was defined as an interval between the J wave of BCG and the foot point of PPG for PTT1, and an interval between the J wave of BCG and main peak of differential PPG as PTT2 [27]. In Figure 4, the R wave of ECG, J wave of BCG, and main peak of differential PPG all have the largest point in one heart-beat cycle (marked by black dots). The foot point of PPG (marked by black dots) is the first zero crossing point of differential PPG [20,28].

### 2.6. PTT Models for Blood Pressure Estimation

PTT is the time interval between the R wave of ECG and a characteristic point of PPG [29]. The commonly used time stamps on the pulse wave are the foot [30,31], peak [32,33] and middle-point of the rising edge [34,35]. In this study, we defined PTT1 as the time interval between the R wave of ECG and foot point of PPG_F_, or the J wave of BCG and foot point of PPG_T_, and PTT2 as the time interval between the R wave of ECG and main peak of differential PPG_F_ (DPGG_F_), or the J wave of BCG and main peak of differential PPG_T_ (DPPG_T_), as shown in Figure 4.

In order to compare the performance of *BP* estimated by reference and the proposed methods, we utilized three *PTT* models to estimate *BP* that had been published previously [36,37,38].
(1)BP=a∗PTT1+b,
(2)BP=a∗ln(PTT1)+b,
(3)BP=a∗PTT1+b∗HR+c,
where *HR* is the heart rate. Moreover, Heard et al. [39] have simplified the pressure–volume distensibility given by Bramwell and Hill equation to propose an inverse square model. Thus, we proposed an inverse model that included *PTT1*, *PTT2* and *HR*,
(4)$$BP=\frac{a}{PTT1}+\frac{b}{PTT2}+c*HR+d.$$

The linear multi-dimension regression model was used to explore the correlation between the independent variable (*x*) and the dependent variable (*y*), and the estimated variable (*ŷ*) [40]. The independent variables include *PPT1*, *PPT2*, and *HR* measured by reference and proposed methods, the dependent variable is the standard blood pressure measured by an electronic blood pressure monitor, and the estimated variable is the output of the model using reference and proposed methods. The regression model is
(5)$$y=B_{0}+B_{1}x_{i1}+B_{2}x_{i2}+\cdots +B_{k}x_{ik}+e_{i},$$
where *e_i_* is the error, and *B* = [*B*_0_, *B*_1_, …, *B_k_*] is the parameter vector of the model. The loss function (root-mean-square error), *E_RMS_*(*B*), is used to evaluate the performance of this method,
(6)$$E_{RMS}\left(B\right)={\left(\frac{1}{n}\sum_{i=1}^{n}{\left(y_{i}-{\hat{y}}_{i}\right)}^{2}\right)}^{0.5},$$
where *n* is the number of data. An optimal estimate of parameter vector was determined from the measurement data by minimizing the loss function [41].

### 2.7. Statistic Analysis

The quantitative data is expressed as the mean ± standard deviation (SD). A two-tailed paired *t*-test is used to compare the average of delay time (DT_ECG-BCG_) between the ECG and BCG, and the delay time (DT_PPGF-PPGT_) between the PPG_F_ and PPG_T_. A *p*-value of 0.05 or less was considered statistically significant. The degree of linear regression between the PTT parameters (*x* variable) by the reference or proposed method and the standard BP (*y* variable) is expressed with a Pearson correlation coefficient, *r*^2^,
(7)$$r^{2}={\left(\frac{n(\sum xy)-(\sum x)\left(\sum y\right)}{\sqrt{\left[n\sum x^{2}-{\left(\sum x\right)}^{2}][n\sum y^{2}-{\left(\sum y\right)}^{2}\right]}}\right)}^{2},$$
where *n* is the number of data. Furthermore, the precision and agreement between the standard BP and estimated BP by reference and proposed methods are compared using a Bland–Altman plot.

### 2.8. Experiment Protocol

Twenty subjects were recruited whose ages were 20.3 ± 1.03 years (eight males and twelve females ranged from 23 to 19 years of age), weights were 55.8 ± 8.86 Kg (from 44 to 75 Kg), and heights were 164.4 ± 7.66 cm (from 152 to 181 cm). This experiment was approved by the Research Ethics Committee of Chung Shan University Hospital (No. CS2-21194), Taichung city, Taiwan.

Subjects were asked to rest for five minutes and fill out an informed consent form before the measurement. The subjects’ information, including age, weight, height, medical treatment of illness, and health condition, was recorded. The subjects suffered from arrhythmia and asthma were excluded from the experiment. An electronic blood pressure monitor (HM-7210, Omron, Osaka City, Japan) was used to measure the blood pressure as the standard in this experiment. The cuff of the blood pressure monitor was wrapped on the left upper arm, the probes of PPG were placed on the first finger of right hand and toes of right foot, separately, and the electrodes of ECG were placed on the chest to measure the Lead I equivalent signal. The subject ran on a treadmill to raise their BP. The experiment procedure is described below:Subjects stood on the weight scale to measure ECG, PPG_F_, PPG_T_, and BCG signals for five minutes, as shown in Figure 5. This measurement was used as the baseline of this experiment. At the same time, the blood pressure was measured once; and its finish-time was marked at the PPG signal.Subjects ran on the treadmill at a speed of about 6 km/h firstly for at least three minutes, and 8 km/h for the last four minutes.Subjects were requested to stand on the weight scale again, to measure ECG, PPG_F_, PPG_T_, and BCG signal for six minutes. The blood pressures of subjects were measured repeatedly once a minute within standing on the weight scale.One measurement would take about 18 min. Each subject was measured four times. The interval between two measurements was at least one week.

Table 1 shows the maximum and minimum standard blood pressures of subjects in the four measurements. Four signals were cut into segments every minute. The qualities of four signals were determined by the manual selection in each segment. If any one of the four signals measured did not have good quality for at least 10 s, the signals within this segment would be ignored. The PTT1 values and PTT2 values were extracted from each heartbeat of four signals, which would be ranked. The average of inter-quartile range of PTT1 values and PTT2 values represented the PTT1 and PTT2 within this one minute. Therefore, the maximum and minimum sample numbers for all subjects were 24 and 12. Table 1 also shows the sample number (N) of each subject.

## 3. Results

In this study, we used four models, Equations (1)–(4), to verify the performance of the proposed method based on the BCG and PPG_T_ signals that was compared with the previous studies as the reference method based on the ECG and PPG_F_ signals [29]. We used *E_RMS_* and correlation coefficient (*r*^2^) to describe the performances of two methods. Table 2 shows the results of estimated systolic and diastolic blood pressures (SBP and DBP) with Equation (1). The *E_RMS_* of estimated SBP and DBP by the reference method are 6.6 ± 2.40 mmHg and 5.1 ± 1.67 mmHg, and *E_RMS_* of estimated SBP and DBP by the proposed method are 9.1 ± 2.14 mmHg and 5.2 ± 1.61 mmHg. The *r*^2^ of reference method for the SBP and DBP are 0.572 ± 0.204 and 0.157 ± 0.128, and *r*^2^ of proposed method for the SBP and DBP are 0.204 ± 0.181 and 0.087 ± 0.111.

Table 3 shows the results of estimated SBP and DBP with Equation (2). The *E_RMS_* of estimated SBP and DBP by the reference method are 6.2 ± 1.74 mmHg and 5.0 ± 1.65 mmHg, and *E_RMS_* of estimated SBP and DBP by the proposed method are 8.6 ± 2.12 mmHg and 5.0 ± 1.48 mmHg. The *r*^2^ of reference method for the SBP and DBP are 0.611 ± 0.157 and 0.167 ± 0.135, and *r*^2^ of proposed method for the SBP and DBP are 0.272 ± 0.241 and 0.119 ± 0.134.

Table 4 shows the results of estimated SBP and DBP with Equation (3). The *E_RMS_* of estimated SBP and DBP by the reference method are 5.7 ± 1.76 mmHg and 4.9 ± 1.72 mmHg, and *E_RMS_* of estimated SBP and DBP by the proposed method are 7.0 ± 1.59 mmHg and 4.8 ± 1.48 mmHg. The *r*^2^ of reference method for the SBP and DBP are 0.699 ± 0.138 and 0.241 ± 0.162, and *r*^2^ of proposed method for the SBP and DBP are 0.548 ± 0.157 and 0.246 ± 0.167.

Table 5 shows the results of estimated SBP and DBP with Equation (4). The *E_RMS_* of estimated SBP and DBP by the reference method are 5.4 ± 1.71 mmHg and 4.9 ± 1.68 mmHg, and *E_RMS_* of estimated SBP and DBP by the proposed method are 6.7 ± 1.60 mmHg and 4.8 ± 1.47 mmHg. The *r*^2^ of reference method for the SBP and DBP are 0.742 ± 0.131 and 0.293 ± 0.169, and *r*^2^ of proposed method for the SBP and DBP are 0.606 ± 0.142 and 0.284 ± 0.166. Figure 6a shows the Bland–Altman plot of estimated SBPs by reference (black) and proposed (red) methods with Equation (4), and the standard SBP. The limits of agreement for the proposed method are larger than the reference method. The SBP differences of proposed method within 100 mmHg to 150 mmHg are larger than the reference method. Figure 6b shows the Bland–Altman plot of estimated DBPs by reference (black) and proposed (red) methods with Equation (4), and the standard DBP. The limits of agreement for the proposed method are smaller than the reference method. The DBP differences of proposed method within the 80 mmHg to 120 mmHg are also smaller than the reference method.

We found that the reference method to estimate BP has better performance than the proposed method. In order to explore this problem, we analyzed the differences of PTT1 and PTT2 measured by reference and proposed methods. Table 6 shows the statistic of total PTT1 and PTT2 that were measured by reference and proposed methods. The number of PTT1 or PTT2 is 396. PTT1_ECG_ is extracted from the ECG and PPG_F_ signals, and PTT1_BCG_ is extracted from the BCG and PPG_T_ signals. PTT2_ECG_ is extracted from the ECG and DPPG signals, and PTT2_BCG_ is extracted from the BCG and DPGG signals. We used the paired *t*-test to compare the differences of PTT1 and PTT2 data between reference and proposed methods. We found that not only is PTT1_ECG_ (164.8 ± 21.46 ms) significantly greater than PTT1_BCG_ (142.2 ± 17.57 ms), but also PTT2_ECG_ (227.4 ± 24.29 ms) is significantly greater than PTT2_BCG_ (212.0 ± 18.81 ms).

The ECG is an electric signal, while BCG is a mechanical signal. Moreover, although PPG_F_ and PPG_T_ are all PPG signals, their transit distances are not the same. Thus, we explored the delay time (DT_ECG-BCG_) between the ECG and BCG, and the delay time (DT_PPGF-PPGT_) between the PPG_F_ and PPG_T_ in Table 7. The number of all pulses used to get the PTT1 and PTT2 is 4364. DT_ECG-BCG_ (82.8 ± 22.73 ms) is significantly greater than DT_PPGF-PPGT_ (61.6 ± 17.47 ms).

## 4. Discussion

The BP is adjusted by the autonomic nervous system and central nervous system. However, in the clinical diagnosis, BP is measured when people have rested for five minutes, which represents the rest BP. Therefore, some studies used the beat-to-beat BP measured by the invasive method to calibrate the BP measured by the cuffless methods [42,43]. Thus, the accuracies of these studies were higher. However, when the blood pressure changes under spinal anesthesia [44] or exercise [12], the accuracy of blood pressure measured by the PTT would become fairly poor. In the pilot study of Sharwood-Smith, the correlation coefficient between the mean blood pressure and PTT was only *r*^2^ = 0.55 [44]. The result for the exercise experiment was only *r*^2^ = 0.49 [12]. In this study, the BP of subjects was raised by the exercise, and measured by the commercial electronic BP monitor. The best *r*^2^ and *E_RMS_* of estimated SBP with the proposed method was 0.606 ± 0.142 and 6.7 ± 1.60 mmHg, respectively. These results were very close to the previous studies.

There are many studies investigating the relations between the PTT and BP [24,26]. The PTT could be defined as the interval between the R wave of ECG and the specific point of PPG because the PTT contains the intra-cardiac (pre-ejection period, PEP) and the vascular transit time [45]. The PET is a time delay that is affected by stress, physical activity, age, and emotion [9]. Thus, the foot point of PPG wave is usually used as the characteristic point to reduce influence of the PEP, which represents the ending time of the cardiac diastolic cycle. Some studies proposed the PTT using the middle-point of the rising edge as the characteristic point of the PPG signal because the rest BP is also controlled by the autonomic nervous system. Therefore, we proposed a new relation between the inverse PTT and BP, as in Equation (4). The PTT includes the PTT1 (interval between the R wave of ECG and the foot point of PPG) and PTT2 (interval between the R wave of ECG and the middle-point of the rising edge of PPG). We found that the proposed model, Equation (4), has the best performance among the other three models, Equations (1)–(3), in terms of *E_RMS_* of estimated SBP: 5.4 ± 1.71 mmHg vs. 6.6 ± 2.40 mmHg, 6.2 ± 1.74 mmHg, and 5.7 ± 1.76 mmHg.

In Table 5, the performance of the proposed method seems not to be better than the reference method in terms of *E_RMS_* of estimated SBP: 6.7 ± 1.60 mmHg vs. 5.4 ± 1.71 mmHg, and *r*^2^ for the SBP with Equation (4): 0.606 ± 0.142 vs. 0.742 ± 0.131. The PTT1 and PTT2 measured by the proposed method are significantly smaller than the reference method in Table 6. Then, we analyzed the delay time between the ECG and BCG, and between the PPGF and PPGT in Table 7. DT_ECG-BCG_ (82.8 ± 22.73 ms) is significantly greater than DT_PPGF-PPGT_ (61.6 ± 17.47 ms). Thus, the accuracy of PTT measured by the BCG is lower than the ECG. Shin et al. used the BCG measured from the wrist and the PPG measured from the finger to estimate the SBP. The correlation coefficient, *r*^2^, approaches 0.64, which is better than 0.606 ± 0.142 in our study [20]. Martin et al. used the BCG and PPG measured from the foot to estimate the SBP [23]. *E_RMS_* was only 11.8 ± 1.6 mm, which was worse than 5.4 ± 1.71 mmHg in our study. Thus, the farther away from the heart the PTT is measured, the less accurate the estimate is.

BCG could be measured when a person is standing on a weight scale [46], lying on a bed [47], or sitting in a chair [48]. The SNR of BCG is affected by the posture of a person [22]. Moreover, the SNR of BCG measured by the weight scale is the lowest among the other two. Inan et al. studied the relation between SNR of BCG and the electromyogram power of feet during movement, and the correlation coefficient, *r*, was 0.89 [46]. Thus, the main limitation of the proposed method is that subjects must stand stably. In this study, subjects were all young people whose leg muscles were normal. That is, the SNR of BCG would be high. If the muscles of users’ legs are abnormal, such as in sarcopenia or handicapped legs, they are not suitable to use this method to measure BP. Moreover, PPG is to measure the change of blood volume in the capillary [49], for which morphologies are different at the different body sites. In Figure 4, the power of PPG measured from the finger is larger than the toe. If users have poor circulation in the lower extremities, the PPG waveform will be indistinct. Thus, the performance of cuffless BP measurement using the weight scale must be worse. However, its advantage is that this method could approach to develop cuffless and touchless BP measurement in the future, when the PPG sensor is embedded in the weight scale.

## 5. Conclusions

Cuffless and touchless BP measurement has the potential benefit of reducing infection risk during the COVID-19 epidemic. In this study, we used the commercial weight scale to sense the BCG signal that replaced the ECG. Four PTT models were used to estimate BP. The PTT of the reference method was detected by the ECG and PPG_F_. The proposed method used the BCG and PPG_T_ to extract the PTT. In terms of the performance of the proposed method, the expected result was worse than that of the reference method. However, *E_RMS_* of estimated SBP by the proposed method is 6.7 ± 1.60 mmHg, which is within the range of agreement. Therefore, this method could be implemented in a cuffless and touchless system to measure the continuous beat-to-beat blood pressure. In the future, we will use deep learning techniques to estimate the BP for increasing the accuracy of blood pressure measurements.

## Figures and Tables

**Figure 1 nutrients-14-02552-f001:**
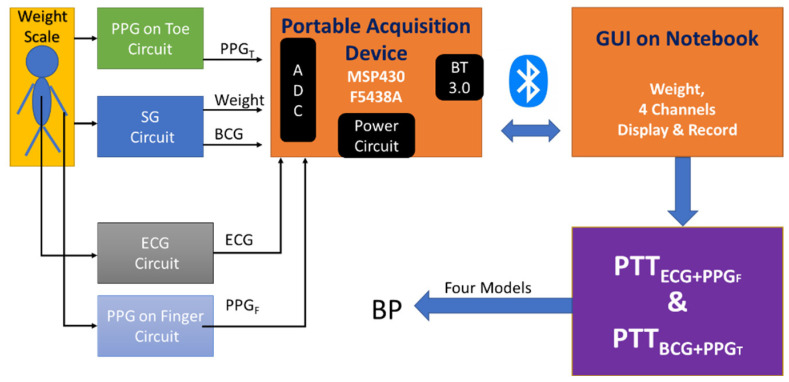
The structure diagram of measurement system including the sensor circuits, portable acquisition device and a graphic user interface (GUI) on the notebook. The PTTECG + PPGF and PTTBCG + PPGT are used to estimate BP with four models. PPG: point of photoplethysmogram; SG: strain gauges; ECG: electrocardiogram; BCG: ballistocardiogram; PTT: pulse transit time; BP: blood pressure.

**Figure 2 nutrients-14-02552-f002:**
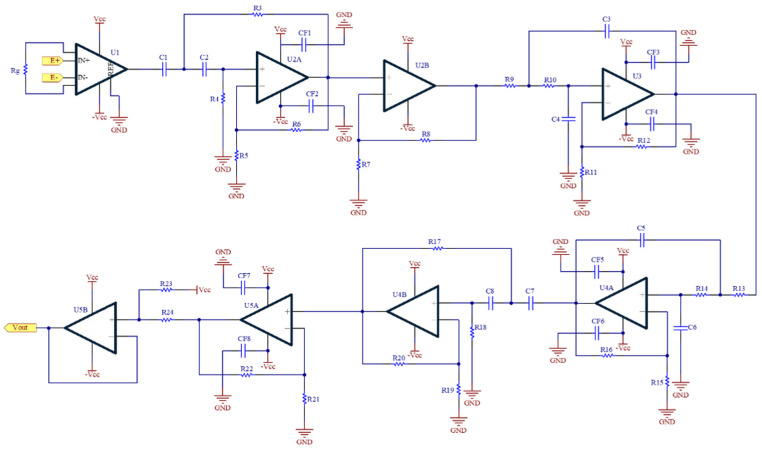
The circuit schematic for BCG measurement.

**Figure 3 nutrients-14-02552-f003:**
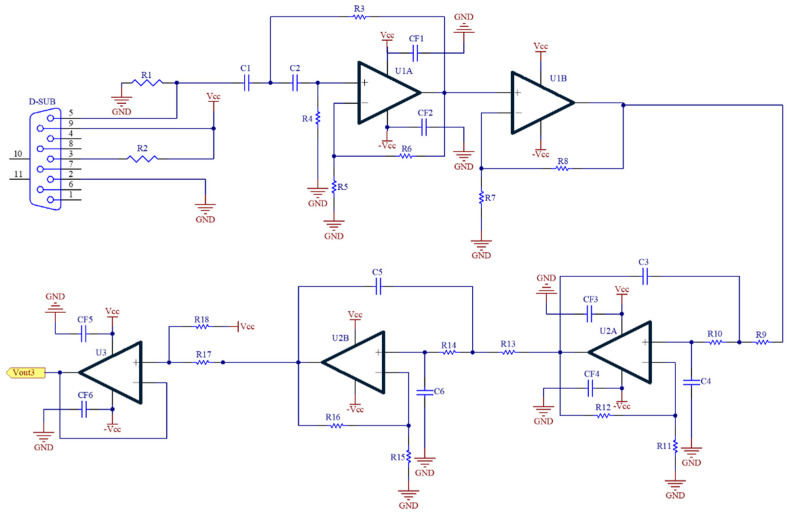
The circuit schematic for PPG measurement.

**Figure 4 nutrients-14-02552-f004:**
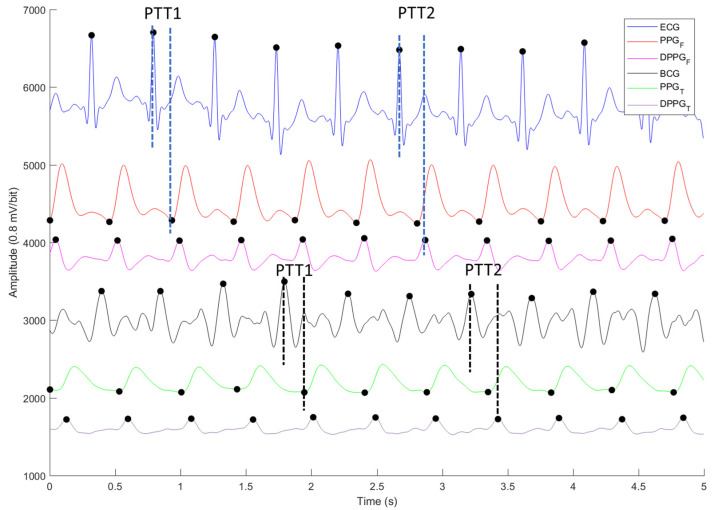
The four signals and their differential signals, ECG (blue), PPGF (red), PPGT (green), BCG (black), differential PPGF (DPPGF, pink) and PPGT (DPPGT, purple). The R wave of ECG, J wave of BCG, main peak of differential PPG, and foot point of PPG are marked by black dots.

**Figure 5 nutrients-14-02552-f005:**
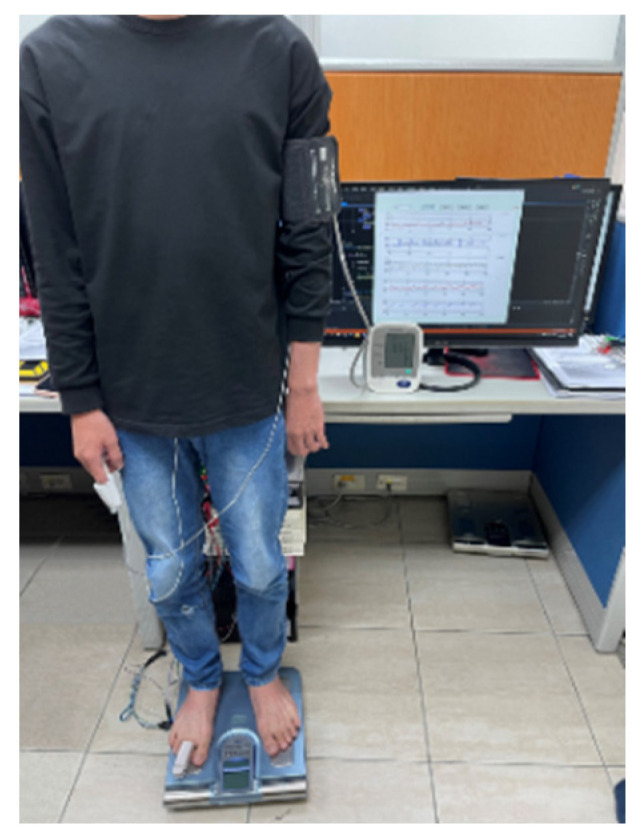
A subject stands on the commercial weight scale. The finger of right hand and toe of right foot wears the PPG probes. The cuff of blood pressure monitor is wrapped on the left hand. Lead I ECG is measured.

**Figure 6 nutrients-14-02552-f006:**
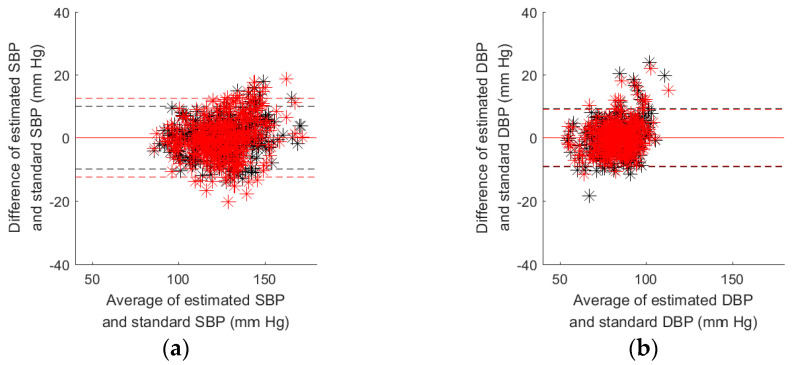
(**a**) Bland–Altman plot of estimated SBPs by reference (black) and proposed (red) methods with Equation (4), and the standard SBP; (**b**) Bland–Altman plot of estimated DBPs by the reference (black) and proposed (red) methods with Equation (4), and the standard SBP.

**Table 1 nutrients-14-02552-t001:** The maximum and minimum standard blood pressures of subjects in the four measurements.

Subject (N)	SBP Max.~Min. mmHg	DBP Max.~Min. mmHg	Subject (N)	SBP Max.~Min. mmHg	DBP Max.~Min. mmHg
01 (N = 21)	142~111	102~74	11 (N = 23)	150~116	106~85
02 (N = 17)	121~84	81~66	12 (N = 24)	154~116	88~58
03 (N = 24)	172~124	121~85	13 (N = 16)	153~126	103~89
04 (N = 24)	156~116	88~68	14 (N = 12)	146~114	77~100
05 (N = 24)	154~106	83~59	15 (N = 12)	132~100	90~76
06 (N = 19)	173~134	104~85	16 (N = 17)	137~104	90~80
07 (N = 23)	152~123	99~83	17 (N = 23)	148~98	90~75
08 (N = 23)	154~118	114~76	18 (N = 16)	113~91	95~67
09 (N = 12)	133~108	85~74	19 (N = 23)	144~112	92~76
10 (N = 24)	132~103	103~74	20 (N = 19)	141~96	74~54

ps: min. is the abbreviation of minimum, max. is the abbreviation of maximum.

**Table 2 nutrients-14-02552-t002:** The results of estimated systolic and diastolic blood pressures with Equation (1).

Subjects	Reference Method				Proposed Method			
	SBP		DBP		SBP		DBP	
	*E_RMS_* (mmHg)	*r* ^2^	*E_RMS_* (mmHg)	*r* ^2^	*E_RMS_* (mmHg)	*r* ^2^	*E_RMS_* (mmHg)	*r* ^2^
01	5.0	0.740	5.2	0.193	9.672	0.031	5.7	0.019
02	6.8	0.472	4.4	0.001	7.598	0.348	4.1	0.145
03	8.2	0.641	7.0	0.062	13.760	0.000	5.7	0.376
04	8.7	0.445	4.9	0.014	11.675	0.005	4.9	0.004
05	4.7	0.831	5.2	0.330	7.608	0.570	5.6	0.234
06	9.6	0.296	4.9	0.098	8.772	0.424	5.1	0.034
07	6.9	0.359	3.8	0.276	8.665	0.012	4.4	0.014
08	7.8	0.476	8.0	0.059	9.609	0.218	8.1	0.030
09	4.6	0.609	2.7	0.023	5.702	0.408	2.7	0.000
10	4.7	0.714	5.8	0.066	6.646	0.443	5.3	0.227
11	5.7	0.583	6.8	0.026	7.624	0.278	6.8	0.000
12	4.8	0.783	6.6	0.159	10.218	0.047	6.4	0.221
13	5.8	0.501	3.5	0.371	8.210	0.003	4.4	0.004
14	5.8	0.685	5.8	0.273	9.523	0.164	6.8	0.003
15	4.1	0.839	3.9	0.155	7.300	0.499	3.7	0.233
16	7.2	0.350	1.9	0.432	8.185	0.175	2.5	0.001
17	13.6	0.063	3.5	0.003	12.846	0.171	3.5	0.008
18	5.1	0.583	7.5	0.143	7.655	0.078	8.1	0.001
19	3.0	0.902	2.7	0.235	9.445	0.039	3.0	0.029
20	9.4	0.565	6.2	0.228	13.079	0.161	6.5	0.155
Mean ± SD	6.6 ± 2.40	0.572 ± 0.204	5.0 ± 1.67	0.157 ± 0.128	9.2 ± 2.14	0.204 ± 0.181	5.2 ± 1.61	0.087 ± 0.111

ps: SD, SBP, DBP are the abbreviations for standard deviation, systolic blood pressure, diastolic blood pressure, respectively.

**Table 3 nutrients-14-02552-t003:** The results of estimated systolic and diastolic blood pressures with Equation (2).

Subjects	Reference Method				Proposed Method			
	SBP		DBP		SBP		DBP	
	*E_RMS_* (mmHg)	*r* ^2^	*E_RMS_* (mmHg)	*r* ^2^	*E_RMS_* (mmHg)	*r* ^2^	*E_RMS_* (mmHg)	*r* ^2^
01	4.8	0.761	5.2	0.178	9.6	0.026	5.8	0.001
02	6.7	0.486	4.4	0.001	6.9	0.458	3.8	0.229
03	8.1	0.650	7.1	0.039	13.2	0.070	6.2	0.256
04	7.9	0.534	4.8	0.029	10.8	0.149	4.8	0.045
05	4.4	0.851	5.0	0.382	6.5	0.684	5.5	0.270
06	9.2	0.354	4.9	0.108	10.8	0.114	5.1	0.016
07	6.9	0.369	3.8	0.276	8.4	0.059	4.3	0.041
08	8.0	0.446	7.7	0.120	8.9	0.326	7.5	0.165
09	5.1	0.511	2.7	0.019	5.1	0.508	2.7	0.025
10	4.9	0.690	5.8	0.067	7.0	0.369	5.5	0.159
11	4.7	0.715	6.6	0.055	8.8	0.034	6.8	0.002
12	6.1	0.659	6.9	0.089	7.1	0.527	5.6	0.393
13	5.5	0.542	3.5	0.382	8.2	0.000	4.4	0.023
14	6.5	0.607	5.7	0.292	9.2	0.207	5.1	0.430
15	3.7	0.870	3.8	0.184	10.2	0.007	4.2	0.026
16	7.0	0.397	1.9	0.439	4.5	0.744	2.5	0.030
17	8.2	0.655	3.5	0.003	8.1	0.666	3.5	0.000
18	5.1	0.590	7.5	0.146	7.5	0.102	8.0	0.039
19	2.8	0.910	2.7	0.205	9.0	0.115	3.0	0.011
20	8.8	0.614	5.8	0.319	12.1	0.274	6.2	0.224
Mean ± SD	6.2 ± 1.74	0.611 ± 0.157	5.0 ± 1.65	0.167 ± 0.135	8.6 ± 2.12	0.272 ± 0.241	5.0 ± 1.48	0.119 ± 0.134

ps: SD, SBP, DBP are the abbreviations for standard deviation, systolic blood pressure, diastolic blood pressure, respectively.

**Table 4 nutrients-14-02552-t004:** The results of estimated systolic and diastolic blood pressures with Equation (3).

Subjects	Reference Method				Proposed Method			
	SBP		DBP		SBP		DBP	
	*E_RMS_* (mmHg)	*r* ^2^	*E_RMS_* (mmHg)	*r* ^2^	*E_RMS_* (mmHg)	*r* ^2^	*E_RMS_* (mmHg)	*r* ^2^
01	4.8	0.766	5.4	0.179	7.3	0.477	5.6	0.112
02	4.3	0.798	3.0	0.553	5.3	0.703	3.4	0.441
03	8.3	0.650	7.2	0.052	11.0	0.384	6.4	0.256
04	7.5	0.608	4.8	0.098	9.0	0.436	4.5	0.210
05	4.5	0.853	4.6	0.505	6.1	0.734	4.1	0.597
06	5.9	0.749	5.0	0.122	6.8	0.673	5.0	0.120
07	7.0	0.378	3.8	0.283	7.6	0.276	4.3	0.102
08	8.0	0.478	7.9	0.121	7.5	0.546	7.6	0.188
09	2.4	0.905	2.7	0.105	4.8	0.607	2.7	0.069
10	5.0	0.693	5.8	0.121	6.9	0.413	5.5	0.197
11	4.3	0.772	6.1	0.243	7.9	0.255	7.0	0.012
12	6.2	0.659	7.0	0.091	6.9	0.579	5.5	0.434
13	5.5	0.570	3.4	0.449	6.3	0.444	3.5	0.407
14	6.8	0.613	6.0	0.294	8.0	0.463	5.4	0.433
15	3.0	0.920	3.7	0.299	6.3	0.662	3.7	0.304
16	5.7	0.618	2.0	0.444	3.5	0.857	2.4	0.158
17	8.3	0.665	3.6	0.013	8.2	0.671	3.6	0.003
18	4.8	0.651	7.8	0.148	4.9	0.644	7.3	0.243
19	2.9	0.911	2.8	0.209	7.4	0.435	2.9	0.126
20	7.8	0.718	5.3	0.482	7.9	0.710	5.2	0.501
Mean ± SD	5.7 ± 1.76	0.699 ± 0.138	4.9 ± 1.72	0.241 ± 0.162	7.0 ± 1.59	0.548 ± 0.157	4.8 ± 1.48	0.246 ± 0.167

ps: SD, SBP, DBP are the abbreviations for standard deviation, systolic blood pressure, diastolic blood pressure, respectively.

**Table 5 nutrients-14-02552-t005:** The results of estimated systolic and diastolic blood pressures with Equation (4).

Subjects	Reference Method				Proposed Method			
	SBP		DBP		SBP		DBP	
	*E_RMS_* (mmHg)	*r* ^2^	*E_RMS_* (mmHg)	*r* ^2^	*E_RMS_* (mmHg)	*r* ^2^	*E_RMS_* (mmHg)	*r* ^2^
01	4.9	0.770	5.4	0.203	5.8	0.684	5.5	0.192
02	4.4	0.807	2.8	0.639	5.2	0.734	3.5	0.447
03	8.3	0.664	6.7	0.219	10.7	0.446	5.7	0.435
04	7.2	0.654	4.8	0.138	8.8	0.483	4.5	0.231
05	4.3	0.871	4.7	0.505	5.8	0.767	4.2	0.615
06	5.9	0.770	5.2	0.121	6.7	0.702	5.2	0.123
07	7.3	0.365	4.0	0.277	7.6	0.305	4.4	0.112
08	7.8	0.534	7.7	0.213	7.6	0.548	7.5	0.255
09	2.6	0.897	2.8	0.107	4.7	0.670	2.9	0.098
10	4.8	0.730	5.9	0.139	6.6	0.491	5.4	0.264
11	4.5	0.772	6.2	0.259	7.3	0.390	7.1	0.014
12	4.8	0.802	6.6	0.247	5.9	0.704	5.8	0.417
13	5.0	0.679	3.4	0.493	5.9	0.548	3.6	0.449
14	4.8	0.830	6.2	0.327	7.8	0.544	5.6	0.449
15	3.1	0.926	3.7	0.386	5.6	0.761	3.9	0.327
16	5.9	0.621	1.9	0.515	3.7	0.851	2.5	0.166
17	7.8	0.722	3.7	0.018	8.4	0.679	3.7	0.008
18	4.1	0.766	8.0	0.173	5.0	0.663	7.5	0.268
19	2.7	0.925	2.7	0.301	7.5	0.441	2.7	0.309
20	7.7	0.738	4.9	0.579	8.0	0.716	5.3	0.508
Mean ± SD	5.4 ± 1.71	0.742 ± 0.131	4.9 ± 1.68	0.293 ± 0.169	6.7 ± 1.60	0.606 ± 0.142	4.8 ± 1.47	0.284 ± 0.166

ps: SD, SBP, DBP are the abbreviations for standard deviation, systolic blood pressure, diastolic blood pressure, respectively.

**Table 6 nutrients-14-02552-t006:** The statistic of total PTT1 and PTT2 measured by reference and proposed methods.

	PTT1_ECG_ (ms)	PTT1_BCG_ (ms)	PTT2_ECG_ (ms)	PTT2_BCG_ (ms)
Mean	164.8	142.2	227.4	212.0
SD	21.46	17.57	24.29	18.81
*p*-value	2.76 × 10^−51^		5.23 × 10^−22^	

ps: SD is the abbreviation of standard deviation.

**Table 7 nutrients-14-02552-t007:** The statistic of delay time (DT_ECG-BCG_) between the ECG and BCG, and the delay time (DT_PPGF-PPGT_) between the PPG_F_ and PPG_T_.

	DT_ECG-BCG_ (ms)	DT_PPG__F__-PPG__T_ (ms)
Mean	82.8	61.6
SD	22.73	17.47
*p*-value	0.000	

ps: SD is the abbreviation of standard deviation.

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
