# Peer review of "Cuffless and Touchless Measurement of Blood Pressure from Ballistocardiogram Based on a Body Weight Scale"

_nutrients, 2022, doi:10.3390/nu14122552_

Round 1

Reviewer 1 Report

This is a very interesting paper describing the design and studies of cuffless device for at home blood pressure measurement. It potentially a very interesting topic with wide audience. However there is no description of statistical analysis package that is used for the study and the description of the devices and procedures need to be clarified so that the reader can understand. 

Please see following are some examples of descriptions that need to be clarified.

The statistical analysis package that were used in the paper need to be described in the methods section. There is no description for  how the r2  was determined in the paper.

--Line 123-124:  “Two PPG probes (DS- 122 100A, Nellcor Puritan Bennett, California, USA) were placed on the middle 123 finger of left hand and toe of left foot, respectively, which sensed the PPG signals, PPGF and PPGT, synchronously.”

Please indicate the sequence of the measurement, which one first, second and last.

--Line  125 : All signals were transferred to a notebook for display and record by the Bluetooth connection.

Would the authors mean: All signals were transferred via Bluetooth connection to a notebook for display and recording.

--Line 127-128: PTTECG+PPGF values were extracted from ECG and PPGF signals, 127 and PTTBCG+PPGT values were extracted from BCG and PPGT signals when sub- 128 ject’s BP being the different levels.

What does the authors meant to say with “subject’s BP being the different levels”

--Line 138:” major part…”

Would the authors mean “Main part”

--Line 139: resolution of analog to digital convert (ADC) was 12 bits.

Would the authors mean “the resolution of analog to digital conversion (ADC) was 12 bits.”

--Line 158:  All filters  are realized with the Butterworth structure.

Would the authors mean” All filters are implemented with Butterworth structure”.

--Line 247: “These signals were”

Would the authors mean” These measurement were…”

--Line 260: Four signals with one minute were segmented.

What is the meaning of this sentence?

--Line 261: “If any one with the good quality of four signals has not at least 10 seconds, the signals within this segment will be ignored.”

Would the authors mean “ If any one of the four signals measured does not have good quality for at least 10 seconds, the signals within this segment would be ignored”

Reviewer 2 Report

The authors explore a creative and convenient method to obtain blood pressure measurements using a body weight scale combined with a photoplethysmogram (PPG) sensor. My comments are as follows:

-          While I agree that this is a cuffless method and that it requires less touching than the standard method, I would not call it touchless because people still need to touch the PPG probe.

-          The introduction mentions several factors that can affect BCG signal and PPG measurements (subject’s skin, tissue, etc.). Based on this, the article needs an expanded limitations section in the discussion, which should mention these technical factors, the relatively small number of subjects, inclusion of only young individuals, etc.

-          The article is well written in general. However, it would benefit from additional English editing, particularly for the excessive use of “the” and some sentences that need revision.

-          Does Table 1 present values from cuff BP measurements?

-          It is not clear how many measurements were included in the multi-dimension regression models: one measurement at the beginning of the experiment and once a minute for a period of 6 minutes after running (so 7 measurements per individual)? Then, this was repeated 3 additional times over several weeks?

-          The “reference” and “proposed” methods are not systematically compared to cuff measurements. As I understand it, you also obtained 7 BP measurements using the cuff method per individual each time. How does the performance of the reference (ECG + PPGf) and proposed (BCG + PPGt) methods compare to the reproducibility of cuff measurements?

-          While I agree that model four performed the best, how do the reference (ECG + PPGf) values correlate with the proposed (BCG + PPGt) values, and with the cuff BP values, including all individuals and all visits?

-          Figure 6. Which model was used for proposed values in this figure? Model 4? Ideally, this figure should present the cuff measurements, the reference method, and the proposed method by model.

Round 2

Reviewer 2 Report

The authors have addressed my concerns and clarified issues related to methodology, limitations, and presentation of findings. It is clear the methodology and model will need to be refined and validated further in the future as it gets tested in more individuals, and particularly, in people with comorbidities. Nonetheless, I think the study has high scientific value. The manuscript still needs some English editing.

Author Response

Reviewer 2 (round 2)
Dear Anonymous Reviewer,
The authors are grateful to your comments and suggestions for improving the quality and presentation of this paper. All comments are followed. The revised parts are highlighted in red. It is our sincere hope that this revision will enhance readability and strengthen of the manuscript to satisfy the requirements of this prestigious journal.

Comments and Suggestions for Authors
The authors have addressed my concerns and clarified issues related to methodology, limitations, and presentation of findings. It is clear the methodology and model will need to be refined and validated further in the future as it gets tested in more individuals, and particularly, in people with comorbidities. Nonetheless, I think the study has high scientific value. The manuscript still needs some English editing.
ANS: Many thanks for reviewer’s comment. The English editing of manuscript has been revised by one (Prof. Wenxi Chen) of the authors who has lived in Japan and other countries for many years.